# Stochastic Expectation Propagation

**Yingzhen Li**
University of Cambridge
Cambridge, CB2 1PZ, UK
yl494@cam.ac.uk

**José Miguel Hernández-Lobato**
Harvard University
Cambridge, MA 02138 USA
jmh@seas.harvard.edu

**Richard E. Turner**
University of Cambridge
Cambridge, CB2 1PZ, UK
ret26@cam.ac.uk

## Abstract

Expectation propagation (EP) is a deterministic approximation algorithm that is often used to perform approximate Bayesian parameter learning. EP approximates the full intractable posterior distribution through a set of local approximations that are iteratively refined for each datapoint. EP can offer analytic and computational advantages over other approximations, such as Variational Inference (VI), and is the method of choice for a number of models. The local nature of EP appears to make it an ideal candidate for performing Bayesian learning on large models in large-scale dataset settings. However, EP has a crucial limitation in this context: the number of approximating factors needs to increase with the number of datapoints, $N$, which often entails a prohibitively large memory overhead. This paper presents an extension to EP, called stochastic expectation propagation (SEP), that maintains a global posterior approximation (like VI) but updates it in a local way (like EP). Experiments on a number of canonical learning problems using synthetic and real-world datasets indicate that SEP performs almost as well as full EP, but reduces the memory consumption by a factor of $N$. SEP is therefore ideally suited to performing approximate Bayesian learning in the large model, large dataset setting.

## 1 Introduction

Recently a number of methods have been developed for applying Bayesian learning to large datasets. Examples include sampling approximations [1, 2], distributional approximations including stochastic variational inference (SVI) [3] and assumed density filtering (ADF) [4], and approaches that mix distributional and sampling approximations [5, 6]. One family of approximation method has garnered less attention in this regard: Expectation Propagation (EP) [7, 8]. EP constructs a posterior approximation by iterating simple local computations that refine factors which approximate the posterior contribution from each datapoint. At first sight, it therefore appears well suited to large-data problems: the locality of computation make the algorithm simple to parallelise and distribute, and good practical performance on a range of small data applications suggest that it will be accurate [9, 10, 11]. However the elegance of local computation has been bought at the price of prohibitive memory overhead that grows with the number of datapoints $N$, since local approximating factors need to be maintained for every datapoint, which typically incur the same memory overhead as the global approximation. The same pathology exists for the broader class of power EP (PEP) algorithms [12] that includes variational message passing [13]. In contrast, variational inference (VI) methods [14, 15] utilise global approximations that are refined directly, which prevents memory overheads from scaling with $N$.

Is there ever a case for preferring EP (or PEP) to VI methods for large data? We believe that there certainly is. First, EP can provide significantly more accurate approximations. It is well known that variational free-energy approaches are biased and often severely so [16] and for particular models the variational free-energy objective is pathologically ill-suited such as those with non-smooth likelihood functions [11, 17]. Second, the fact that EP is truly local (to factors in the posterior distri-

bution and not just likelihoods) means that it affords different opportunities for tractable algorithm design, as the updates can be simpler to approximate.

As EP appears to be the method of choice for some applications, researchers have attempted to push it to scale. One approach is to swallow the large computational burden and simply use large data structures to store the approximating factors (e.g. TrueSkill [18]). This approach can only be pushed so far. A second approach is to use ADF, a simple variant of EP that only requires a global approximation to be maintained in memory [19]. ADF, however, provides poorly calibrated uncertainty estimates [7] which was one of the main motivating reasons for developing EP in the first place. A third idea, complementary to the one described here, is to use approximating factors that have simpler structure (e.g. low rank, [20]). This reduces memory consumption (e.g. for Gaussian factors from $\mathcal{O}(ND^2)$ to $\mathcal{O}(ND)$), but does not stop the scaling with $N$. Another idea uses EP to carve up the dataset [5, 6] using approximating factors for collections of datapoints. This results in coarse-grained, rather than local, updates and other methods must be used to compute them. (Indeed, the spirit of [5, 6] is to extend sampling methods to large datasets, not EP itself.)

Can we have the best of both worlds? That is, accurate global approximations that are derived from truly local computation. To address this question we develop an algorithm based upon the standard EP and ADF algorithms that maintains a global approximation which is updated in a local way. We call this class of algorithms Stochastic Expectation Propagation (SEP) since it updates the global approximation with (damped) stochastic estimates on data sub-samples in an analogous way to SVI. Indeed, the generalisation of the algorithm to the PEP setting directly relates to SVI. Importantly, SEP reduces the memory footprint by a factor of $N$ when compared to EP. We further extend the method to control the granularity of the approximation, and to treat models with latent variables without compromising on accuracy or unnecessary memory demands. Finally, we demonstrate the scalability and accuracy of the method on a number of real world and synthetic datasets.

## 2    Expectation Propagation and Assumed Density Filtering

We begin by briefly reviewing the EP and ADF algorithms upon which our new method is based. Consider for simplicity observing a dataset comprising $N$ i.i.d. samples $\mathcal{D} = \{\boldsymbol{x}_n\}_{n=1}^N$ from a probabilistic model $p(\boldsymbol{x}|\boldsymbol{\theta})$ parametrised by an unknown $D$-dimensional vector $\boldsymbol{\theta}$ that is drawn from a prior $p_0(\boldsymbol{\theta})$. Exact Bayesian inference involves computing the (typically intractable) posterior distribution of the parameters given the data,

$$p(\boldsymbol{\theta}|\mathcal{D}) \propto p_0(\boldsymbol{\theta}) \prod_{n=1}^N p(\boldsymbol{x}_n|\boldsymbol{\theta}) \approx q(\boldsymbol{\theta}) \propto p_0(\boldsymbol{\theta}) \prod_{n=1}^N f_n(\boldsymbol{\theta}). \tag{1}$$

Here $q(\boldsymbol{\theta})$ is a simpler tractable approximating distribution that will be refined by EP. The goal of EP is to refine the approximate factors so that they capture the contribution of each of the likelihood terms to the posterior i.e. $f_n(\boldsymbol{\theta}) \approx p(\boldsymbol{x}_n|\boldsymbol{\theta})$. In this spirit, one approach would be to find each approximating factor $f_n(\boldsymbol{\theta})$ by minimising the Kullback-Leibler (KL) divergence between the posterior and the distribution formed by replacing one of the likelihoods by its corresponding approximating factor, $\mathrm{KL}[p(\boldsymbol{\theta}|\mathcal{D})||p(\boldsymbol{\theta}|\mathcal{D})f_n(\boldsymbol{\theta})/p(\boldsymbol{x}_n|\boldsymbol{\theta})]$. Unfortunately, such an update is still intractable as it involves computing the full posterior. Instead, EP approximates this procedure by replacing the exact leave-one-out posterior $p_{-n}(\boldsymbol{\theta}) \propto p(\boldsymbol{\theta}|\mathcal{D})/p(\boldsymbol{x}_n|\boldsymbol{\theta})$ on both sides of the KL by the approximate leave-one-out posterior (called the cavity distribution) $q_{-n}(\boldsymbol{\theta}) \propto q(\boldsymbol{\theta})/f_n(\boldsymbol{\theta})$. Since this couples the updates for the approximating factors, the updates must now be iterated.

In more detail, EP iterates four simple steps. First, the factor selected for update is removed from the approximation to produce the cavity distribution. Second, the corresponding likelihood is included to produce the tilted distribution $\tilde{p}_n(\boldsymbol{\theta}) \propto q_{-n}(\boldsymbol{\theta})p(\boldsymbol{x}_n|\boldsymbol{\theta})$. Third EP updates the approximating factor by minimising $\mathrm{KL}[\tilde{p}_n(\boldsymbol{\theta})||q_{-n}(\boldsymbol{\theta})f_n(\boldsymbol{\theta})]$. The hope is that the contribution the true-likelihood makes to the posterior is similar to the effect the same likelihood has on the tilted distribution. If the approximating distribution is in the exponential family, as is often the case, then the KL minimisation reduces to a moment matching step [21] that we denote $f_n(\boldsymbol{\theta}) \leftarrow \mathtt{proj}[\tilde{p}_n(\boldsymbol{\theta})]/q_{-n}(\boldsymbol{\theta})$. Finally, having updated the factor, it is included into the approximating distribution.

We summarise the update procedure for a single factor in Algorithm 1. Critically, the approximation step of EP involves local computations since one likelihood term is treated at a time. The assumption

| **Algorithm 1** EP | **Algorithm 2** ADF | **Algorithm 3** SEP |
|---|---|---|
| 1: choose a factor $f_n$ to refine: | 1: choose a datapoint $\boldsymbol{x}_n \sim \mathcal{D}$: | 1: choose a datapoint $\boldsymbol{x}_n \sim \mathcal{D}$: |
| 2: compute cavity distribution $q_{-n}(\boldsymbol{\theta}) \propto q(\boldsymbol{\theta})/f_n(\boldsymbol{\theta})$ | 2: compute cavity distribution $q_{-n}(\boldsymbol{\theta}) = q(\boldsymbol{\theta})$ | 2: compute cavity distribution $q_{-1}(\boldsymbol{\theta}) \propto q(\boldsymbol{\theta})/f(\boldsymbol{\theta})$ |
| 3: compute tilted distribution $\tilde{p}_n(\boldsymbol{\theta}) \propto p(\boldsymbol{x}_n|\boldsymbol{\theta})q_{-n}(\boldsymbol{\theta})$ | 3: compute tilted distribution $\tilde{p}_n(\boldsymbol{\theta}) \propto p(\boldsymbol{x}_n|\boldsymbol{\theta})q_{-n}(\boldsymbol{\theta})$ | 3: compute tilted distribution $\tilde{p}_n(\boldsymbol{\theta}) \propto p(\boldsymbol{x}_n|\boldsymbol{\theta})q_{-1}(\boldsymbol{\theta})$ |
| 4: moment matching: $f_n(\boldsymbol{\theta}) \leftarrow \mathtt{proj}[\tilde{p}_n(\boldsymbol{\theta})]/q_{-n}(\boldsymbol{\theta})$ | 4: moment matching: $f_n(\boldsymbol{\theta}) \leftarrow \mathtt{proj}[\tilde{p}_n(\boldsymbol{\theta})]/q_{-n}(\boldsymbol{\theta})$ | 4: moment matching: $f_n(\boldsymbol{\theta}) \leftarrow \mathtt{proj}[\tilde{p}_n(\boldsymbol{\theta})]/q_{-1}(\boldsymbol{\theta})$ |
| 5: inclusion: $q(\boldsymbol{\theta}) \leftarrow q_{-n}(\boldsymbol{\theta})f_n(\boldsymbol{\theta})$ | 5: inclusion: $q(\boldsymbol{\theta}) \leftarrow q_{-n}(\boldsymbol{\theta})f_n(\boldsymbol{\theta})$ | 5: inclusion: $q(\boldsymbol{\theta}) \leftarrow q_{-1}(\boldsymbol{\theta})f_n(\boldsymbol{\theta})$ |
| | | 6: *implicit update*: $f(\boldsymbol{\theta}) \leftarrow f(\boldsymbol{\theta})^{1-\frac{1}{N}} f_n(\boldsymbol{\theta})^{\frac{1}{N}}$ |

Figure 1: Comparing the Expectation Propagation (EP), Assumed Density Filtering (ADF), and Stochastic Expectation Propagation (SEP) update steps. Typically, the algorithms will be initialised using $q(\boldsymbol{\theta}) = p_0(\boldsymbol{\theta})$ and, where appropriate, $f_n(\boldsymbol{\theta}) = 1$ or $f(\boldsymbol{\theta}) = 1$.

is that these local computations, although possibly requiring further approximation, are far simpler to handle compared to the full posterior $p(\boldsymbol{\theta}|\mathcal{D})$. In practice, EP often performs well when the updates are parallelised. Moreover, by using approximating factors for groups of datapoints, and then running additional approximate inference algorithms to perform the EP updates (which could include nesting EP), EP carves up the data making it suitable for distributed approximate inference.

There is, however, one wrinkle that complicates deployment of EP at scale. Computation of the cavity distribution requires removal of the current approximating factor, which means any implementation of EP must store them explicitly necessitating an $\mathcal{O}(N)$ memory footprint. One option is to simply ignore the removal step replacing the cavity distribution with the full approximation, resulting in the ADF algorithm (Algorithm 2) that needs only maintain a global approximation in memory. But as the moment matching step now over-counts the underlying approximating factor (consider the new form of the objective $\mathrm{KL}[q(\boldsymbol{\theta})p(\boldsymbol{x}_n|\boldsymbol{\theta})||q(\boldsymbol{\theta})]$) the variance of the approximation shrinks to zero as multiple passes are made through the dataset. Early stopping is therefore required to prevent overfitting and generally speaking ADF does not return uncertainties that are well-calibrated to the posterior. In the next section we introduce a new algorithm that sidesteps EP's large memory demands whilst avoiding the pathological behaviour of ADF.

## 3   Stochastic Expectation Propagation

In this section we introduce a new algorithm, inspired by EP, called Stochastic Expectation Propagation (SEP) that combines the benefits of local approximation (tractability of updates, distributability, and parallelisability) with global approximation (reduced memory demands). The algorithm can be interpreted as a version of EP in which the approximating factors are tied, or alternatively as a corrected version of ADF that prevents overfitting. The key idea is that, at convergence, the approximating factors in EP can be interpreted as parameterising a global factor, $f(\boldsymbol{\theta})$, that captures the average effect of a likelihood on the posterior $f(\boldsymbol{\theta})^N \triangleq \prod_{n=1}^{N} f_n(\boldsymbol{\theta}) \approx \prod_{n=1}^{N} p(\boldsymbol{x}_n|\boldsymbol{\theta})$. In this spirit, the new algorithm employs direct iterative refinement of a global approximation comprising the prior and $N$ copies of a single approximating factor, $f(\boldsymbol{\theta})$, that is $q(\boldsymbol{\theta}) \propto f(\boldsymbol{\theta})^N p_0(\boldsymbol{\theta})$.

SEP uses updates that are analogous to EP's in order to refine $f(\boldsymbol{\theta})$ in such a way that it captures the average effect a likelihood function has on the posterior. First the cavity distribution is formed by removing one of the copies of the factor, $q_{-1}(\boldsymbol{\theta}) \propto q(\boldsymbol{\theta})/f(\boldsymbol{\theta})$. Second, the corresponding likelihood is included to produce the tilted distribution $\tilde{p}_n(\boldsymbol{\theta}) \propto q_{-1}(\boldsymbol{\theta})p(\boldsymbol{x}_n|\boldsymbol{\theta})$ and, third, SEP finds an intermediate factor approximation by moment matching, $f_n(\boldsymbol{\theta}) \leftarrow \mathtt{proj}[\tilde{p}_n(\boldsymbol{\theta})]/q_{-1}(\boldsymbol{\theta})$. Finally, having updated the factor, it is included into the approximating distribution. It is important here not to make a full update since $f_n(\boldsymbol{\theta})$ captures the effect of just a single likelihood function $p(\boldsymbol{x}_n|\boldsymbol{\theta})$. Instead, damping should be employed to make a partial update $f(\boldsymbol{\theta}) \leftarrow f(\boldsymbol{\theta})^{1-\epsilon} f_n(\boldsymbol{\theta})^{\epsilon}$. A natural choice uses $\epsilon = 1/N$ which can be interpreted as minimising $\mathrm{KL}[\tilde{p}_n(\boldsymbol{\theta})||p_0(\boldsymbol{\theta})f(\boldsymbol{\theta})^N]$

in the moment update, but other choices of $\epsilon$ may be more appropriate, including decreasing $\epsilon$ according to the Robbins-Monro condition [22].

SEP is summarised in Algorithm 3. Unlike ADF, the cavity is formed by dividing out $f(\boldsymbol{\theta})$ which captures the average affect of the likelihood and prevents the posterior from collapsing. Like ADF, however, SEP only maintains the global approximation $q(\boldsymbol{\theta})$ since $f(\boldsymbol{\theta}) \propto (q(\boldsymbol{\theta})/p_0(\boldsymbol{\theta}))^{\frac{1}{N}}$ and $q_{-1}(\boldsymbol{\theta}) \propto q(\boldsymbol{\theta})^{1-\frac{1}{N}} p_0(\boldsymbol{\theta})^{\frac{1}{N}}$. When Gaussian approximating factors are used, for example, SEP reduces the storage requirement of EP from $\mathcal{O}(ND^2)$ to $\mathcal{O}(D^2)$ which is a substantial saving that enables models with many parameters to be applied to large datasets.

# 4 Algorithmic extensions to SEP and theoretical results

SEP has been motivated from a practical perspective by the limitations inherent in EP and ADF. In this section we extend SEP in four orthogonal directions relate SEP to SVI. Many of the algorithms described here are summarised in Figure 2 and they are detailed in the supplementary material.

## 4.1 Parallel SEP: relating the EP fixed points to SEP

The SEP algorithm outlined above approximates one likelihood at a time which can be computationally slow. However, it is simple to parallelise the SEP updates by following the same recipe by which EP is parallelised. Consider a minibatch comprising $M$ datapoints (for a full parallel batch update use $M = N$). First we form the cavity distribution for each likelihood. Unlike EP these are all identical. Next, in parallel, compute $M$ intermediate factors $f_m(\boldsymbol{\theta}) \leftarrow \texttt{proj}[\tilde{p}_m(\boldsymbol{\theta})]/q_{-1}(\boldsymbol{\theta})$. In EP these intermediate factors become the new likelihood approximations and the approximation is updated to $q(\boldsymbol{\theta}) = p_0(\boldsymbol{\theta}) \prod_{n \neq m} f_n(\boldsymbol{\theta}) \prod_m f_m(\boldsymbol{\theta})$. In SEP, the same update is used for the approximating distribution, which becomes $q(\boldsymbol{\theta}) \leftarrow p_0(\boldsymbol{\theta}) f_{old}(\boldsymbol{\theta})^{N-M} \prod_m f_m(\boldsymbol{\theta})$ and, by implication, the approximating factor is $f_{new}(\boldsymbol{\theta}) = f_{old}(\boldsymbol{\theta})^{1-M/N} \prod_{m=1}^{M} f_m(\boldsymbol{\theta})^{1/N}$. One way of understanding parallel SEP is as a double loop algorithm. The **inner loop** produces intermediate approximations $q_m(\boldsymbol{\theta}) \leftarrow \arg\min_q \text{KL}[\tilde{p}_m(\boldsymbol{\theta})||q(\boldsymbol{\theta})]$; these are then combined in the **outer loop**: $q(\boldsymbol{\theta}) \leftarrow \arg\min_q \sum_{m=1}^{M} \text{KL}[q(\boldsymbol{\theta})||q_m(\boldsymbol{\theta})] + (N - M)\text{KL}[q(\boldsymbol{\theta})||q_{old}(\boldsymbol{\theta})]$.

For $M = 1$ parallel SEP reduces to the original SEP algorithm. For $M = N$ parallel SEP is equivalent to the so-called Averaged EP algorithm proposed in [23] as a theoretical tool to study the convergence properties of normal EP. This work showed that, under fairly restrictive conditions (likelihood functions that are log-concave and varying slowly as a function of the parameters), AEP converges to the same fixed points as EP in the large data limit ($N \to \infty$).

There is another illuminating connection between SEP and AEP. Since SEP's approximating factor $f(\boldsymbol{\theta})$ converges to the geometric average of the intermediate factors $\bar{f}(\boldsymbol{\theta}) \propto [\prod_{n=1}^{N} f_n(\boldsymbol{\theta})]^{\frac{1}{N}}$, SEP converges to the same fixed points as AEP if the learning rates satisfy the Robbins-Monro condition [22], and therefore under certain conditions [23], to the same fixed points as EP. But it is still an open question whether there are more direct relationships between EP and SEP.

## 4.2 Stochastic power EP: relationships to variational methods

The relationship between variational inference and stochastic variational inference [3] mirrors the relationship between EP and SEP. Can these relationships be made more formal? If the moment projection step in EP is replaced by a natural parameter matching step then the resulting algorithm is equivalent to the Variational Message Passing (VMP) algorithm [24] (and see supplementary material). Moreover, VMP has the same fixed points as variational inference [13] (since minimising the local variational KL divergences is equivalent to minimising the global variational KL).

These results carry over to the new algorithms with minor modifications. Specifically VMP can be transformed into SVMP by replacing VMP's local approximations with the global form employed by SEP. In the supplementary material we show that this algorithm is an instance of standard SVI and that it therefore has the same fixed points as VI when $\epsilon$ satisfies the Robbins-Monro condition [22]. More generally, the procedure can be applied any member of the power EP (PEP) [12] family of algorithms which replace the moment projection step in EP with alpha-divergence minimization

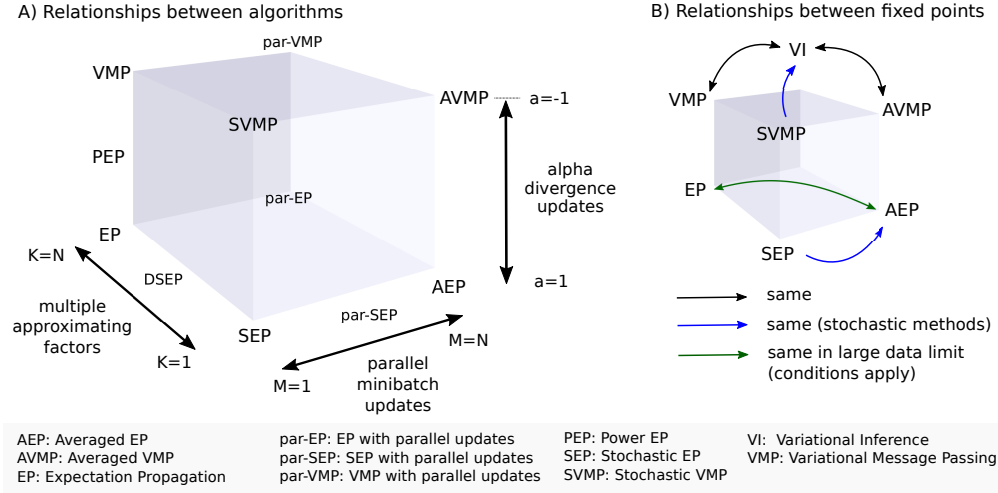

Figure 2: Relationships between algorithms. Note that care needs to be taken when interpreting the alpha-divergence as $a \to -1$ (see supplementary material).

[21], but care has to be taken when taking the limiting cases (see supplementary). These results lend weight to the view that SEP is a natural stochastic generalisation of EP.

### 4.3 Distributed SEP: controlling granularity of the approximation

EP uses a fine-grained approximation comprising a single factor for each likelihood. SEP, on the other hand, uses a coarse-grained approximation comprising a signal global factor to approximate the average effect of all likelihood terms. One might worry that SEP's approximation is too severe if the dataset contains sets of datapoints that have very different likelihood contributions (e.g. for odd-vs-even handwritten digits classification consider the affect of a 5 and a 9 on the posterior). It might be more sensible in such cases to partition the dataset into $K$ disjoint pieces $\{\mathcal{D}_k = \{\boldsymbol{x}_n\}_{n=N_{k-1}}^{N_k}\}_{k=1}^K$ with $N = \sum_{k=1}^K N_k$ and use an approximating factor for each partition. If normal EP updates are performed **on** the subsets, i.e. treating $p(\mathcal{D}_k|\boldsymbol{\theta})$ as a single true factor to be approximated, we arrive at the Distributed EP algorithm [5, 6]. But such updates are challenging as multiple likelihood terms must be included during each update necessitating additional approximations (e.g. MCMC). A simpler alternative uses SEP/AEP **inside** each partition, implying a posterior approximation of the form $q(\boldsymbol{\theta}) \propto p_0(\boldsymbol{\theta}) \prod_{k=1}^K f_k(\boldsymbol{\theta})^{N_k}$ with $f_k(\boldsymbol{\theta})^{N_k}$ approximating $p(\mathcal{D}_k|\boldsymbol{\theta})$. The limiting cases of this algorithm, when $K = 1$ and $K = N$, recover SEP and EP respectively.

### 4.4 SEP with latent variables

Many applications of EP involve latent variable models. Although this is not the main focus of the paper, we show that SEP is applicable in this case without scaling the memory footprint with $N$. Consider a model containing hidden variables, $\boldsymbol{h}_n$, associated with each observation $p(\boldsymbol{x}_n, \boldsymbol{h}_n|\boldsymbol{\theta})$ that are drawn i.i.d. from a prior $p_0(\boldsymbol{h}_n)$. The goal is to approximate the true posterior over parameters and hidden variables $p(\boldsymbol{\theta}, \{\boldsymbol{h}_n\}|\mathcal{D}) \propto p_0(\boldsymbol{\theta}) \prod_n p_0(\boldsymbol{h}_n) p(\boldsymbol{x}_n|\boldsymbol{h}_n, \boldsymbol{\theta})$. Typically, EP would approximate the effect of each intractable term as $p(\boldsymbol{x}_n|\boldsymbol{h}_n, \boldsymbol{\theta}) p_0(\boldsymbol{h}_n) \approx f_n(\boldsymbol{\theta}) g_n(\boldsymbol{h}_n)$. Instead, SEP ties the approximate parameter factors $p(\boldsymbol{x}_n|\boldsymbol{h}_n, \boldsymbol{\theta}) p_0(\boldsymbol{h}_n) \approx f(\boldsymbol{\theta}) g_n(\boldsymbol{h}_n)$ yielding:

$$q(\boldsymbol{\theta}, \{\boldsymbol{h}_n\}) \stackrel{\triangle}{\propto} p_0(\boldsymbol{\theta}) f(\boldsymbol{\theta})^N \prod_{n=1}^N g_n(\boldsymbol{h}_n). \tag{2}$$

Critically, as proved in supplementary, the local factors $g_n(\boldsymbol{h}_n)$ do not need to be maintained in memory. This means that all of the advantages of SEP carry over to more complex models involving latent variables, although this can potentially increase computation time in cases where updates for $g_n(\boldsymbol{h}_n)$ are not analytic, since then they will be initialised from scratch at each update.

# 5 Experiments

The purpose of the experiments was to evaluate SEP on a number of datasets (synthetic and real-world, small and large) and on a number of models (probit regression, mixture of Gaussians and Bayesian neural networks).

## 5.1 Bayesian probit regression

The first experiments considered a simple Bayesian classification problem and investigated the stability and quality of SEP in relation to EP and ADF as well as the effect of using mini-batches and varying the granularity of the approximation. The model comprised a probit likelihood function $P(\boldsymbol{y}_n = 1|\theta) = \Phi(\boldsymbol{\theta}^T \boldsymbol{x}_n)$ and a Gaussian prior over the hyper-plane parameter $p(\boldsymbol{\theta}) = \mathcal{N}(\boldsymbol{\theta}; \mathbf{0}, \gamma I)$. The synthetic data comprised $N = 5,000$ datapoints $\{(\boldsymbol{x}_n, \boldsymbol{y}_n)\}$, where $\boldsymbol{x}_n$ were $D = 4$ dimensional and were either sampled from a single Gaussian distribution (Fig. 3(a)) or from a mixture of Gaussians (MoGs) with $J = 5$ components (Fig. 3(b)) to investigate the sensitivity of the methods to the homogeneity of the dataset. The labels were produced by sampling from the generative model. We followed [6] measuring the performance by computing an approximation of $\mathrm{KL}[p(\boldsymbol{\theta}|\mathcal{D})||q(\boldsymbol{\theta})]$, where $p(\boldsymbol{\theta}|\mathcal{D})$ was replaced by a Gaussian that had the same mean and covariance as samples drawn from the posterior using the No-U-Turn sampler (NUTS) [25], to quantify the calibration of uncertainty estimations.

Results in Fig. 3(a) indicate that EP is the best performing method and that ADF collapses towards a delta function. SEP converges to a solution which appears to be of similar quality to that obtained by EP for the dataset containing Gaussian inputs, but slightly worse when the MoGs was used. Variants of SEP that used larger mini-batches fluctuated less, but typically took longer to converge (although for the small minibatches shown this effect is not clear). The utility of finer grained approximations depended on the homogeneity of the data. For the second dataset containing MoG inputs (shown in Fig. 3(b)), finer-grained approximations were found to be advantageous if the datapoints from each mixture component are assigned to the same approximating factor. Generally it was found that there is no advantage to retaining more approximating factors than there were clusters in the dataset.

To verify whether these conclusions about the granularity of the approximation hold in real datasets, we sampled $N = 1,000$ datapoints for each of the digits in MNIST and performed odd-vs-even classification. Each digit class was assigned its own global approximating factor, $K = 10$. We compare the log-likelihood of a test set using ADF, SEP ($K = 1$), full EP and DSEP ($K = 10$) in Figure 3(c). EP and DSEP significantly outperform ADF. DSEP is slightly worse than full EP initially, however it reduces the memory to 0.001% of full EP without losing accuracy substantially. SEP's accuracy was still increasing at the end of learning and was slightly better than ADF. Further empirical comparisons are reported in the supplementary, and in summary the three EP methods are indistinguishable when likelihood functions have similar contributions to the posterior.

Finally, we tested SEP's performance on six small binary classification datasets from the UCI machine learning repository.[1] We did not consider the effect of mini-batches or the granularity of the approximation, using $K = M = 1$. We ran the tests with damping and stopped learning after convergence (by monitoring the updates of approximating factors). The classification results are summarised in Table 1. ADF performs reasonably well on the mean classification error metric, presumably because it tends to learn a good approximation to the posterior mode. However, the posterior variance is poorly approximated and therefore ADF returns poor test log-likelihood scores. EP achieves significantly higher test log-likelihood than ADF indicating that a superior approximation to the posterior variance is attained. Crucially, SEP performs very similarly to EP, implying that SEP is an accurate alternative to EP even though it is refining a cheaper global posterior approximation.

## 5.2 Mixture of Gaussians for clustering

The small scale experiments on probit regression indicate that SEP performs well for fully-observed probabilistic models. Although it is not the main focus of the paper, we sought to test the flexibility of the method by applying it to a latent variable model, specifically a mixture of Gaussians. A synthetic MoGs dataset containing $N = 200$ datapoints was constructed comprising $J = 4$ Gaussians.

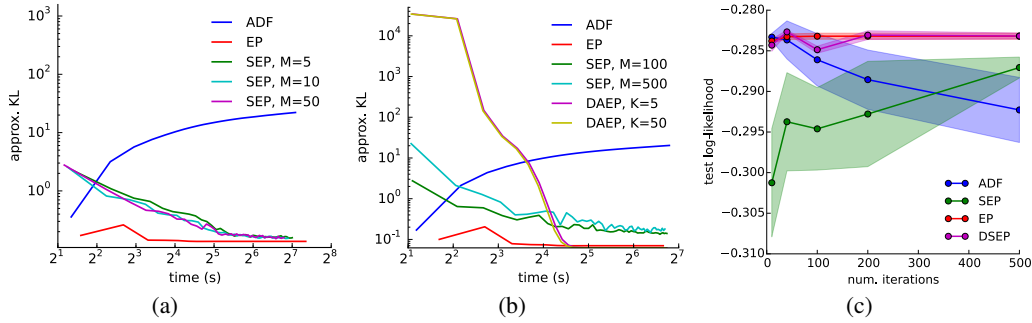

Figure 3: Bayesian logistic regression experiments. Panels (a) and (b) show synthetic data experiments. Panel (c) shows the results on MNIST (see text for full details).

Table 1: Average test results all methods on probit regression. All methods appear to capture the posterior's mode, however EP outperforms ADF in terms of test log-likelihood on almost all of the datasets, with SEP performing similarly to EP.

| Dataset | mean error | | | test log-likelihood | | |
| | ADF | SEP | EP | ADF | SEP | EP |
|---|---|---|---|---|---|---|
| Australian | 0.328±0.0127 | **0.325±0.0135** | 0.330±0.0133 | -0.634±0.010 | -0.631±0.009 | **-0.631±0.009** |
| Breast | 0.037±0.0045 | **0.034±0.0034** | 0.034±0.0039 | -0.100±0.015 | -0.094±0.011 | **-0.093±0.011** |
| Crabs | 0.056±0.0133 | **0.033±0.0099** | 0.036±0.0113 | -0.242±0.012 | -0.125±0.013 | **-0.110±0.013** |
| Ionos | **0.126±0.0166** | 0.130±0.0147 | 0.131±0.0149 | -0.373±0.047 | -0.336±0.029 | **-0.324±0.028** |
| Pima | 0.242±0.0093 | 0.244±0.0098 | **0.241±0.0093** | -0.516±0.013 | -0.514±0.012 | **-0.513±0.012** |
| Sonar | **0.198±0.0208** | 0.198±0.0217 | 0.198±0.0243 | -0.461±0.053 | -0.418±0.021 | **-0.415±0.021** |

The means were sampled from a Gaussian distribution, $p(\boldsymbol{\mu}_j) = \mathcal{N}(\boldsymbol{\mu}; \boldsymbol{m}, \mathrm{I})$, the cluster identity variables were sampled from a uniform categorical distribution $p(\boldsymbol{h}_n = j) = 1/4$, and each mixture component was isotropic $p(\boldsymbol{x}_n|\boldsymbol{h}_n) = \mathcal{N}(\boldsymbol{x}_n; \boldsymbol{\mu}_{\boldsymbol{h}_n}, 0.5^2 I)$. EP, ADF and SEP were performed to approximate the joint posterior over the cluster means $\{\boldsymbol{\mu}_j\}$ and cluster identity variables $\{\boldsymbol{h}_n\}$ (the other parameters were assumed known).

Figure 4(a) visualises the approximate posteriors after 200 iterations. All methods return good estimates for the means, but ADF collapses towards a point estimate as expected. SEP, in contrast, captures the uncertainty and returns nearly identical approximations to EP. The accuracy of the methods is quantified in Fig. 4(b) by comparing the approximate posteriors to those obtained from NUTS. In this case the approximate KL-divergence measure is analytically intractable, instead we used the averaged F-norm of the difference of the Gaussian parameters fitted by NUTS and EP methods. These measures confirm that SEP approximates EP well in this case.

## 5.3 Probabilistic backpropagation

The final set of tests consider more complicated models and large datasets. Specifically we evaluate the methods for probabilistic backpropagation (PBP) [4], a recent state-of-the-art method for scalable Bayesian learning in neural network models. Previous implementations of PBP perform several iterations of ADF over the training data. The moment matching operations required by ADF are themselves intractable and they are approximated by first propagating the uncertainty on the synaptic weights forward through the network in a sequential way, and then computing the gradient of the marginal likelihood by backpropagation. ADF is used to reduce the large memory cost that would be required by EP when the amount of available data is very large.

We performed several experiments to assess the accuracy of different implementations of PBP based on ADF, SEP and EP on regression datasets following the same experimental protocol as in [4] (see supplementary material). We considered neural networks with 50 hidden units (except for *Year* and *Protein* which we used 100). Table 2 shows the average test RMSE and test log-likelihood for each method. Interestingly, SEP can outperform EP in this setting (possibly because the stochasticity enabled it to find better solutions), and typically it performed similarly. Memory reductions using

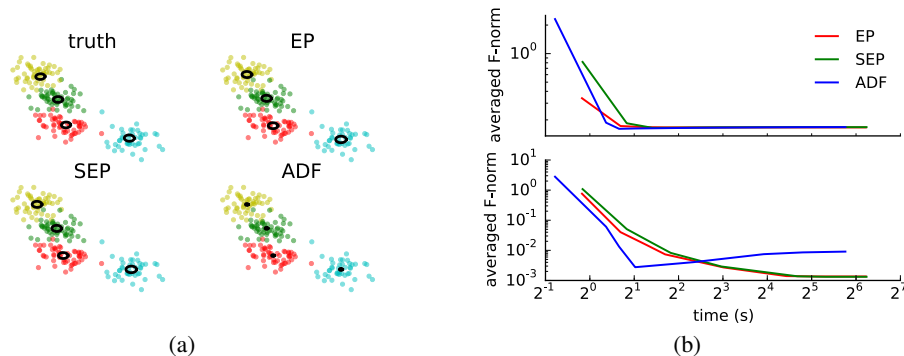

(a)　　　　　　　　　　　　　　　(b)

Figure 4: Posterior approximation for the mean of the Gaussian components. (a) visualises posterior approximations over the cluster means (98% confidence level). The coloured dots indicate the true label (top-left) or the inferred cluster assignments (the rest). In (b) we show the error (in F-norm) of the approximate Gaussians' means (top) and covariances (bottom).

Table 2: Average test results for all methods. Datasets are also from the UCI machine learning repository.

| | RMSE | | | test log-likelihood | | |
|---|---|---|---|---|---|---|
| **Dataset** | **ADF** | **SEP** | **EP** | **ADF** | **SEP** | **EP** |
| Kin8nm | 0.098±0.0007 | **0.088±0.0009** | 0.089±0.0006 | 0.896±0.006 | **1.013±0.011** | 1.005±0.007 |
| Naval | 0.006±0.0000 | **0.002±0.0000** | 0.004±0.0000 | 3.731±0.006 | **4.590±0.014** | 4.207±0.011 |
| Power | **4.124±0.0345** | 4.165±0.0336 | 4.191±0.0349 | **-2.837±0.009** | -2.846±0.008 | -2.852±0.008 |
| Protein | 4.727±0.0112 | **4.670±0.0109** | 4.748±0.0137 | -2.973±0.003 | **-2.961±0.003** | -2.979±0.003 |
| Wine | **0.635±0.0079** | 0.650±0.0082 | 0.637±0.0076 | -0.968±0.014 | -0.976±0.013 | **-0.958±0.011** |
| Year | **8.879± NA** | 8.922±NA | 8.914±NA | **-3.603± NA** | -3.924±NA | -3.929±NA |

SEP instead of EP were large e.g. 694Mb for the Protein dataset and 65,107Mb for the Year dataset (see supplementary). Surprisingly ADF often outperformed EP, although the results presented for ADF use a near-optimal number of sweeps and further iterations generally degraded performance. ADF's good performance is most likely due to an interaction with additional moment approximation required in PBP that is more accurate as the number of factors increases.

# 6　Conclusions and future work

This paper has presented the stochastic expectation propagation method for reducing EP's large memory consumption which is prohibitive for large datasets. We have connected the new algorithm to a number of existing methods including assumed density filtering, variational message passing, variational inference, stochastic variational inference and averaged EP. Experiments on Bayesian logistic regression (both synthetic and real world) and mixture of Gaussians clustering indicated that the new method had an accuracy that was competitive with EP. Experiments on the probabilistic back-propagation on large real world regression datasets again showed that SEP comparably to EP with a vastly reduced memory footprint. Future experimental work will focus on developing data-partitioning methods to leverage finer-grained approximations (DESP) that showed promising experimental performance and also mini-batch updates. There is also a need for further theoretical understanding of these algorithms, and indeed EP itself. Theoretical work will study the convergence properties of the new algorithms for which we only have limited results at present. Systematic comparisons of EP-like algorithms and variational methods will guide practitioners to choosing the appropriate scheme for their application.

**Acknowledgements**

We thank the reviewers for valuable comments. YL thanks the Schlumberger Foundation Faculty for the Future fellowship on supporting her PhD study. JMHL acknowledges support from the Rafael del Pino Foundation. RET thanks EPSRC grant # EP/G050821/1 and EP/L000776/1.

## Footnotes

[1] https://archive.ics.uci.edu/ml/index.html

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
