[Supplementary Material · nips_supp.pdf]

# Stochastic Expectation Propagation: Supplementary Material

**Yingzhen Li**
University of Cambridge
Cambridge, CB2 1PZ, UK
yl494@cam.ac.uk

**José Miguel Hernández-Lobato**
Harvard University
Cambridge, MA 02138 USA
jmh@seas.harvard.edu

**Richard E. Turner**
University of Cambridge
Cambridge, CB2 1PZ, UK
ret26@cam.ac.uk

The supplementary material is divided into these sections. Section A details the design of stochastic power EP methods and presents relationships between SEP and SVI. Section B extends the discussion of distributed algorithms and SEP's applicability to latent variable models. Section C provides experimental details of the Bayesian neural network experiments and presents further emprical evalucations of the method.

## A  Further theoretical connections

We described the extensions of stochastic expectation propagation (SEP) in the main text, and we provide more details in this section.

### A.1  Power EP and alpha-EP

The relationship between EP and variational inference (VI) can be explained by introducing power EP (PEP) [1]. As a preparation let us consider the alpha-divergence[1] introduced in [2]

$$D_\alpha[p(\boldsymbol{\theta})||q(\boldsymbol{\theta})] = \frac{4}{1-\alpha^2}\left(1 - \int_{\boldsymbol{\theta}} p(\boldsymbol{\theta})^{\frac{1+\alpha}{2}} q(\boldsymbol{\theta})^{\frac{1-\alpha}{2}} d\boldsymbol{\theta}\right). \tag{1}$$

Two cases of KL-divergence also belongs to the family of alpha-divergence by definition:

$$D_1[p(\boldsymbol{\theta})||q(\boldsymbol{\theta})] \stackrel{\triangle}{=} \lim_{\alpha \to 1} D_\alpha[p(\boldsymbol{\theta})||q(\boldsymbol{\theta})] = \mathrm{KL}[p(\boldsymbol{\theta})||q(\boldsymbol{\theta})], \tag{2}$$

$$D_{\text{-}1}[p(\boldsymbol{\theta})||q(\boldsymbol{\theta})] \stackrel{\triangle}{=} \lim_{\alpha \to \text{-}1} D_\alpha[p(\boldsymbol{\theta})||q(\boldsymbol{\theta})] = \mathrm{KL}[q(\boldsymbol{\theta})||p(\boldsymbol{\theta})]. \tag{3}$$

Minka [1] also introduced alpha-EP as a generalisation of EP to alpha-divergences, which changes the moment matching step to alpha-projection [3] that returns the minimiser of the alpha divergence $D_\alpha[\tilde{p}_n(\boldsymbol{\theta})||q(\boldsymbol{\theta})]$ wrt. $q(\boldsymbol{\theta})$. Examples include moment projection `proj`[·] which takes $\alpha = 1$, and information projection which chooses $\alpha = -1$. However alpha-projections are difficult to compute in general, motivating power EP (Algorithm 1) – so called because it raises potentials to a power before referencing standard EP updates – as a practical alternative. Minka [1] showed that power EP with power $1/\beta, \beta < \infty$ returns a local optimum of the alpha divergence with $\alpha = -1 + 2/\beta$ when converged. However this still leaves the pathological case $\alpha = -1$ or $\beta = \infty$ since the above equivalence does not apply. Thus variational message passing (VMP), which takes $\alpha \to -1$, cannot be interpreted as a special case of power EP. This observation extends to stochastic PEP as well (Algorithm 2). Instead we derive stochastic VMP in the spirit which SEP extends EP, which keeps the computational steps using current global estimate but ties all the local factors. We discuss this extension in detail in the next section and provide its connection to stochastic variational inference.

**Algorithm 1** PEP

1: choose a factor $f_n$ to refine:
2: compute cavity distribution
    $q_{-n}(\boldsymbol{\theta}) \propto q(\boldsymbol{\theta})/f_n(\boldsymbol{\theta})^{1/\beta}$
3: compute tilted distribution
    $\tilde{p}_n(\boldsymbol{\theta}) \propto p(\boldsymbol{x}_n|\boldsymbol{\theta})^{1/\beta}q_{-n}(\boldsymbol{\theta})$
4: moment matching:
    $f_n(\boldsymbol{\theta}) \leftarrow [\texttt{proj}[\tilde{p}_n(\boldsymbol{\theta})]/q_{-n}(\boldsymbol{\theta})]^{\beta}$
5: inclusion:
    $q(\boldsymbol{\theta}) \leftarrow q(\boldsymbol{\theta})f_n(\boldsymbol{\theta})/f_n^{old}(\boldsymbol{\theta})$

**Algorithm 2** Stochastic PEP

1: choose a datapoint $\boldsymbol{x}_n \sim \mathcal{D}$:
2: compute cavity distribution
    $q_{-1}(\boldsymbol{\theta}) \propto q(\boldsymbol{\theta})/f(\boldsymbol{\theta})^{1/\beta}$
3: compute tilted distribution
    $\tilde{p}_n(\boldsymbol{\theta}) \propto p(\boldsymbol{x}_n|\boldsymbol{\theta})^{1/\beta}q_{-1}(\boldsymbol{\theta})$
4: moment matching:
    $f_n(\boldsymbol{\theta}) \leftarrow [\texttt{proj}[\tilde{p}_n(\boldsymbol{\theta})]/q_{-1}(\boldsymbol{\theta})]^{\beta}$
5: inclusion:
    $q(\boldsymbol{\theta}) \leftarrow q(\boldsymbol{\theta})f_n(\boldsymbol{\theta})/f(\boldsymbol{\theta})$
6: *implicit update*:
    $f(\boldsymbol{\theta}) \leftarrow f(\boldsymbol{\theta})^{1-\frac{1}{N}}f_n(\boldsymbol{\theta})^{\frac{1}{N}}$

### A.2 Connecting SVMP to SVI

We first briefly sketch the VMP algorithm using the EP framework, but replacing the moment matching step with natural parameter matching. We assume the approximate posterior $q(\boldsymbol{\theta})$ is in some exponential family:

$$q(\boldsymbol{\theta}) \propto \exp\left[\langle \boldsymbol{\lambda}_q, \boldsymbol{\phi}(\boldsymbol{\theta})\rangle\right]. \tag{4}$$

At time $t$ we have the current estimate of the natural parameter $\boldsymbol{\lambda}_q^t$, which is defined as the sum of local variational parameters[2]: $\boldsymbol{\lambda}_q^t \triangleq \boldsymbol{\lambda}_0 + \sum_{n=1}^N \boldsymbol{\lambda}_n^t$. Here $\boldsymbol{\lambda}_0$ represents the natural parameter of the prior distribution $p_0(\boldsymbol{\theta})$. VMP iteratively computes the update of each local estimate $\boldsymbol{\lambda}_n^{t+1}$ in the following procedure. First VMP computes the expected sufficient statistics $\hat{\boldsymbol{s}}_n$ about datapoint $\boldsymbol{x}_n$ using $\boldsymbol{\lambda}_q^t$, e.g. $\hat{\boldsymbol{s}}_n = E_q[t(z_n, x_n)]$ in the original SVI paper [4]. Then VMP forms the gradient as though optimising the maximised evidence lower bound (ELBO) but with $q_{-1}(\boldsymbol{\theta})$ as the prior:

$$\nabla_{\boldsymbol{\lambda}_q^t}\mathcal{L} = \boldsymbol{\lambda}_{-1}^t + \hat{\boldsymbol{s}}_n - \boldsymbol{\lambda}_q^t, \tag{5}$$

$$\boldsymbol{\lambda}_{-1}^t = \boldsymbol{\lambda}_q^t - \boldsymbol{\lambda}_n^t. \tag{6}$$

Next VMP zeros the gradient and recovers the current update $\boldsymbol{\lambda}_n^{t+1} = \hat{\boldsymbol{s}}_n$. The stochastic version of VMP, if extended in a way as SEP developed from EP, defines the global variational parameters as $\boldsymbol{\lambda}_q^t \triangleq \boldsymbol{\lambda}_0 + N\boldsymbol{\lambda}^t$. It computes the expected sufficient statistics $\hat{\boldsymbol{s}}_n$ in the same way as VMP but changes the cavity to $\boldsymbol{\lambda}_{-1}^t = \boldsymbol{\lambda}_q^t - \boldsymbol{\lambda}^t$ in the ELBO maximisation steps. Readers can verify that this returns the current update $\boldsymbol{\lambda}^{t+1} = \hat{\boldsymbol{s}}_n$ using the important fact that the local update ONLY depends on the global parameter $\boldsymbol{\lambda}_q^t$. Now since we tie all the local updates, the global parameter update $\boldsymbol{\lambda}_q^{t+1} = \boldsymbol{\lambda}_0 + N\boldsymbol{\lambda}^{t+1} = \boldsymbol{\lambda}_0 + N\hat{\boldsymbol{s}}_n$. In practice we perform a damped update, where a typical choice of step size is $\epsilon = 1/N$ like in SEP:

$$\boldsymbol{\lambda}_q^{t+1} \leftarrow (1 - \frac{1}{N})\boldsymbol{\lambda}_q^t + \frac{1}{N}(\boldsymbol{\lambda}_0 + N\boldsymbol{\lambda}^{t+1}) = \boldsymbol{\lambda}_0 + (N-1)\boldsymbol{\lambda}^t + \hat{\boldsymbol{s}}_n. \tag{7}$$

On the other hand, [5] summarises stochastic variational inference (SVI) as to compute the current update by zeroing the gradient

$$\nabla_{\boldsymbol{\lambda}_q}\mathcal{L} = \boldsymbol{\lambda}_0 + N\hat{\boldsymbol{s}}_n - \boldsymbol{\lambda}_q, \tag{8}$$

which returns $\boldsymbol{\lambda}_q^{t+1} = \boldsymbol{\lambda}_0 + N\hat{\boldsymbol{s}}_n$ as well. This implies that SVI, when using learning rate $\epsilon = 1/N$, is equivalent to SVMP.

### A.3 SEP from a global approximation perspective

In this section we provide some intuition about SEP via an interpretation as approximating minimisation of a global divergence like VI (although it is computed in a truly local way). This framework utilises alpha-divergence, but on the global posterior, and we motivate it by describing VI and SVI as

Figure 1: (a) A geometric view of AEP/PEP comparison. (b) A cartoon illustration of DEP, SEP and DSEP. For each algorithm we show the approximate posterior on the top and the tilted distribution at the bottom.

divergence minimisation. VI performs global optimisation on $\text{KL}[q(\boldsymbol{\theta})||p(\boldsymbol{\theta}|\mathcal{D})]$, and its stochastic version, SVI, can be interpreted as at each step minimising $\text{KL}[q(\boldsymbol{\theta})||p(\boldsymbol{\theta}|\{\boldsymbol{x}_n\}^N)]$ with the $N$ replicas $\{\boldsymbol{x}_n\}^N = \{\boldsymbol{x}_n, \boldsymbol{x}_n, ..., \boldsymbol{x}_n\}$. Similarly, we state SEP as a stochastic global optimisation procedure, which computes an iterative procedure to minimise alpha-divergence $D_\alpha[p(\boldsymbol{\theta}|\{\boldsymbol{x}_n\}^N)||q(\boldsymbol{\theta})]$ with $\alpha = -1 + 2/N$. Indeed we can understand the inner-loop of AEP as PEP with power $1/N$ if considering $f(\boldsymbol{\theta})^N$ as a large composite factor to approximate the likelihood term of $\boldsymbol{x}_n$ raised to power $N$.

However minimising the alpha-divergence between the true posterior $p(\boldsymbol{\theta}|\mathcal{D})$ and the global approximation $q(\boldsymbol{\theta})$ recovers PEP on the whole dataset instead, and the factor to include in the tilted distribution changes to the intractable geometric average $\text{avg}[\{p(\boldsymbol{x}_n|\boldsymbol{\theta})\}] \overset{\triangle}{=} [\prod_n p(\boldsymbol{x}_n|\boldsymbol{\theta})]^{1/N}$. Readers might have noticed that the update of PEP on the full dataset is given by $q(\boldsymbol{\theta}) \leftarrow \text{proj}[\text{avg}[\{\tilde{p}_n(\boldsymbol{\theta})\}]]$. In other words, we can interpret AEP as an approximation to the impractical batch PEP by interchanging projections and averaging operations, and we illustrate a geometric view for this in Fig. 1(a).

It is important to note that SEP/AEP at convergence does not minimise the alpha divergence globally. Like PEP, the inner-loop computes $\text{proj}[\tilde{p}_n(\boldsymbol{\theta})]$, where one can show that it moves towards minimising $D_\alpha(p(\boldsymbol{\theta}|\{\boldsymbol{x}_n\}^N)||q_n(\boldsymbol{\theta}))$ using the same techniques as before. However the outer-loop averages the natural parameters of the intermediate answers, which does not follow the gradient direction of alpha-divergence minimisation. Furthermore, local/global optimisation of alpha divergence are inconsistent in terms of fixed points except at $\alpha = -1$, the divergence utilised in VI and VMP. Indeed we provide the fixed point conditions of AEP which reveals its local nature.

**Proposition 1.** *The fixed points of averaged EP, if they exist, can be written as* $q(\boldsymbol{\theta}) = \text{avg}[\{q_n(\boldsymbol{\theta})\}]$, *where*

$$q_n(\boldsymbol{\theta}) = \text{proj}[\tilde{p}_n(\boldsymbol{\theta})], \tag{9}$$

$$\tilde{p}_n(\boldsymbol{\theta}) \propto q(\boldsymbol{\theta})\frac{p(\boldsymbol{x}_n|\boldsymbol{\theta})}{f(\boldsymbol{\theta})}. \tag{10}$$

*These fixed points are also the fixed points of stochastic EP when the learning rates satisfy the Robbins-Monro condition [6].*

This fixed point condition applies to stochastic PEP as well when $\alpha \neq -1$, and importantly it also implies the pathology of constructing SVMP by using SPEP and limiting $\alpha$ to $-1$.

# B  Algorithmic design details

## B.1  Distributed computing methods

We have shown in the main paper that a proper design of data partitioning improves SEP's approximation accuracy. This distributed algorithm is inspired by the Distributed EP (DEP) algorithm [7, 8]

| **Algorithm 3** DEP | **Algorithm 4** DSEP | **Algorithm 5** DAEP |
|---|---|---|
| 1: compute cavity distribution $q_{-k}(\boldsymbol{\theta}) \propto q(\boldsymbol{\theta})/f_k(\boldsymbol{\theta})$ | 1: compute cavity distribution $q_{-1}(\boldsymbol{\theta}) = q(\boldsymbol{\theta})/f_k(\boldsymbol{\theta})$ | 1: compute cavity distribution $q_{-1}(\boldsymbol{\theta}) \propto q(\boldsymbol{\theta})/f_k(\boldsymbol{\theta})$ |
| 2: compute tilted distribution $\tilde{p}_k(\boldsymbol{\theta}) \propto p(\mathcal{D}_k|\boldsymbol{\theta})q_{-k}(\boldsymbol{\theta})$ | 2: choose a datapoint $\boldsymbol{x}_n \sim \mathcal{D}_k$ | 2: for each $\boldsymbol{x}_n \in \mathcal{D}_k$: |
| 3: moment matching: $f_k(\boldsymbol{\theta}) \leftarrow \mathtt{proj}[\tilde{p}_k(\boldsymbol{\theta})]/q_{-k}(\boldsymbol{\theta})$ | 3: compute tilted distribution $\tilde{p}_k^n(\boldsymbol{\theta}) \propto p(\boldsymbol{x}_n|\boldsymbol{\theta})q_{-1}(\boldsymbol{\theta})$ | 3:    compute tilted distribution $\tilde{p}_k^n(\boldsymbol{\theta}) \propto p(\boldsymbol{x}_n|\boldsymbol{\theta})q_{-1}(\boldsymbol{\theta})$ |
| | 4: moment matching: $f_k^n(\boldsymbol{\theta}) \leftarrow \mathtt{proj}[\tilde{p}_k^n(\boldsymbol{\theta})]/q_{-1}(\boldsymbol{\theta})$ | 4:    moment matching: $f_k^n(\boldsymbol{\theta}) \leftarrow \mathtt{proj}[\tilde{p}_k^n(\boldsymbol{\theta})]/q_{-1}(\boldsymbol{\theta})$ |
| | 5: inclusion: $f_k(\boldsymbol{\theta}) \leftarrow f_k(\boldsymbol{\theta})^{1-1/N_k} f_k^n(\boldsymbol{\theta})^{1/N_k}$ | 5: inclusion: $f_k(\boldsymbol{\theta})^{N_k} \leftarrow \prod_n f_k^n(\boldsymbol{\theta})$ |

Figure 2: Comparing the variants of distributed design for Expectation Propagation (EP) on the current data piece $\mathcal{D}_k$. One should notice that the definitions of $f_k(\boldsymbol{\theta})$ are different for DEP and DSEP/DAEP. Distributed EP (DEP) uses sampling methods to compute the projection step, while Distributed SEP/AEP (DSEP/DAEP) still keeps deterministic computations.

presented in Algorithm 3. DEP first partitions the dataset into $K$ disjoint pieces $\{D_k = \{\boldsymbol{x}_i\}_{i=1}^{N_k}\}$ with $N = \sum_{i=1}^K N_k$, which is well-justified since the true posterior can also be derived as

$$p(\boldsymbol{\theta}|\mathcal{D}) \propto p_0(\boldsymbol{\theta}) \prod_{k=1}^K p(\mathcal{D}_k|\boldsymbol{\theta}), \tag{11}$$

$$p(\mathcal{D}_k|\boldsymbol{\theta}) = \prod_{\boldsymbol{x}_n \in \mathcal{D}_k} p(\boldsymbol{x}_n|\boldsymbol{\theta}). \tag{12}$$

Next DEP assigns factors to each sub-dataset likelihood, i.e. $q(\boldsymbol{\theta}) \propto p_0(\boldsymbol{\theta}) \prod_k f_k(\boldsymbol{\theta})$ with each $f_k(\boldsymbol{\theta})$ approximating $p(\mathcal{D}_k|\boldsymbol{\theta})$. The projection step is no longer analytically tractable in general since the tilted distribution with multiple datapoints often lacks a simple form. Instead DEP handles moment matching with sampling, making it stochastic in the sense of having an stochastic approximation of the moments.

To construct a deterministic counterpart of DEP, we consider running SEP/AEP **inside** each partition. We name this approach as Distributed SEP/AEP (DSEP/DAEP) and provide a comparison in Fig. 1(b) with DEP and SEP **on** the sub-dataset likelihood factors using sampling protocol. Different from DEP, the approximate posterior for DSEP is defined as $q(\boldsymbol{\theta}) \propto p_0(\boldsymbol{\theta}) \prod_k f_k(\boldsymbol{\theta})^{N_k}$, with $f_k(\boldsymbol{\theta})^{N_k}$ approximating $p(\mathcal{D}_k|\boldsymbol{\theta})$. The computations are almost the same as SEP/AEP except that the updates only modify the copies of the corresponded subset. These two algorithms are also detailed in Algorithm 4 and 5, respectively. In section C.3 we provide an emprical study on comparing SEP, EP and DSEP approximations.

## B.2 SEP with latent variables

In this section we show the applicability of SEP to latent variables without scaling the memory consumption with $N$. We consider a model containing latent variables $\boldsymbol{h}_n$ associated with each observation $\boldsymbol{x}_n$, which are drawn i.i.d. from a prior $p_0(\boldsymbol{h}_n)$. SEP proposes approximations to the true posterior over parameters and hidden variables

$$p(\boldsymbol{\theta}, \{\boldsymbol{h}_n\}|\mathcal{D}) \propto p_0(\boldsymbol{\theta}) \prod_n p_0(\boldsymbol{h}_n)p(\boldsymbol{x}_n|\boldsymbol{h}_n, \boldsymbol{\theta}) \tag{13}$$

by tying the factors for the global parameter $\boldsymbol{\theta}$ but retaining the local factors for the hidden variables:

$$q(\boldsymbol{\theta}, \{\boldsymbol{h}_n\}) \stackrel{\triangle}{\propto} p_0(\boldsymbol{\theta})f(\boldsymbol{\theta})^N \prod_{n=1}^N g_n(\boldsymbol{h}_n). \tag{14}$$

In other words, SEP uses $f(\boldsymbol{\theta})g_n(\boldsymbol{h}_n)$ to approximate $p(\boldsymbol{x}_n|\boldsymbol{h}_n, \boldsymbol{\theta})p_0(\boldsymbol{h}_n)$.

Next we show a critical advantage of SEP when approximating the latent variable posterior distributions: the local factors $g_n(\boldsymbol{h}_n)$ do not need to be maintained in memory (though see caveats mentioned below). More formally, the cavity distribution is $q_{-n}(\boldsymbol{\theta}, \{\boldsymbol{h}_n\}) \propto q(\boldsymbol{\theta}, \{\boldsymbol{h}_n\})/(f(\boldsymbol{\theta})g_n(\boldsymbol{h}_n))$

Table 1: Datasets Used in the Experiments with Neural Networks. The memory numbers reported include dataset storage and temporal maintainance of computation graphs in Theano ($\sim 100MB$).

| Dataset | $N$ | $D$ | MB (EP) | MB (SEP) | MB reduced |
|---|---|---|---|---|---|
| Kin8nm | 8192 | 8 | 168.23 | 109.76 | 58.47 |
| Naval Propulsion | 11,934 | 16 | 261.75 | 113.92 | 147.83 |
| Combined Cycle Power Plant | 9568 | 4 | 148.70 | 110.99 | 37.71 |
| Protein Structure | 45,730 | 9 | 815.55 | 121.52 | 694.02 |
| Wine Quality Red | 1599 | 11 | 122.21 | 107.90 | 14.30 |
| **Year Prediction MSD** | **515,345** | **90** | **67837.90** | **2730.55** | **65107.34** |

and the tilted distribution is $\tilde{p}_n(\boldsymbol{\theta}, \{\boldsymbol{h}_n\}) \propto q_{-n}(\boldsymbol{\theta}, \{\boldsymbol{h}_n\})p(\boldsymbol{x}_n|\boldsymbol{h}_n,\boldsymbol{\theta})p_0(\boldsymbol{h}_n)$. This leads to the a moment-update that minimises

$$\mathrm{KL}\left[p_0(\boldsymbol{\theta})f(\boldsymbol{\theta})^{N-1}p(\boldsymbol{x}_n|\boldsymbol{h}_n,\boldsymbol{\theta})p_0(\boldsymbol{h}_n)\prod_{m \neq n}g_m(\boldsymbol{h}_m)||p_0(\boldsymbol{\theta})f(\boldsymbol{\theta})^{N-1}f'(\boldsymbol{\theta})g_n(\boldsymbol{h}_n)\prod_{m \neq n}g_m(\boldsymbol{h}_m)\right].$$

with respect to $f'(\boldsymbol{\theta})g_n(\boldsymbol{h}_n)$. Importantly, the terms involving $\prod_{m \neq n}g_m(\boldsymbol{h}_m)$ cancel, meaning that these factors do not contribute to the local approximation step. For simple models the moments of $\boldsymbol{h}_n$ can be computed analytically given $q_{-1}(\boldsymbol{\theta})$, thus $g_n(\boldsymbol{h}_n)$ is never stored in memory resulting in a reduced memory footprint by a factor of $N$ again. However in practice people may prefer maintaining the $g$ factors in memory, if the moment computation requires another optimisation inner-loop (which might be more expensive than the moment matching step itself). Examples include latent Dirichlet allocation [9] that has a hierachy of latent variables, where VI methods also store variational $q$ distributions for some of the hidden variables. One potential recipe in this scenario is to learn the moments/messages passed in each SEP step in the spirit of [10, 11].

It is also possible to have latent variables globally shared or shared in a data subset $\mathcal{D}_k$. But we can also extend SEP to these latent variables accordingly, which still provides computation gains in space complexity. In mathematical forms, assume $\boldsymbol{h}_k$ a latent variable shared in $\mathcal{D}_k$. Then we construct $q(\boldsymbol{h}_k) \propto p_0(\boldsymbol{h}_k)g_k(\boldsymbol{h}_k)^{N_k}$ to approximate its posterior. This procedure still reduces memory by a factor of $N/K$.

## C Further experimental results

### C.1 Details of Bayesian neural network tests

We perform neural network regression experiments with publicly available data sets and neural networks with one hidden layer. Table 1 lists the analyzed data sets and shows summary statistics. We use neural networks with 50 hidden units in all cases except in the two largest ones, i.e., *Year Prediction MSD* and *Protein Structure*, where we use 100 hidden units. The different methods, SEP, EP and ADF were run by performing 40 passes over the available training data, updating the parameters of the posterior approximation after seeing each data point. The data sets are split into random training and test sets with 90% and 10% of the data, respectively. This splitting process is repeated 20 times and the average test performance of each method is reported. In the two largest data sets, *Year Prediction MSD* and *Protein Structure*, we do the train-test splitting only one and five times respectively. The data sets are normalized so that the input features and the targets have zero mean and unit variance in the training set. The normalization on the targets is removed for prediction.

We also provide the memory consumption details for experiments using probabilistic back-propagation (PBP) in Table 1. We observe substantial memory reductions by running SEP instead of EP, while still attaining similar accuracies. Especially for Year Prediction MSD dataset, which is a typical large-scale dataset both in the number of observations $N$ and the dimensionality $D$, SEP achieves saving tens of gigabytes. We performed the test for EP using a machine with more than 100GB RAM, while SEP only required 2.7GB memory, including the space of storing the dataset (1.9GB). These numbers reveal the huge memory requirement of full EP and further support SEP as a practical alternative in big data, big model settings.

Figure 3: Performance of EP methods on Bayesian logistic regression with sampling moment computations, measured in approximate KL divergence described in the main text.

## C.2 Stochastic EP with sampling protocal

Although not a main purpose, we further test the performance of SEP when using sampling methods to compute moments [3]. We re-use the settings of probit regression but change the probit unit to sigmoid function, making the moment projection analytically intractable. We randomly partition the dataset into $K = 20$ subsets $\{\mathcal{D}_k\}$, construct the approximate posterior with local factors over the subsets, and tie them in SEP/AEP as before. Note that we perform sequential computations for DEP and AEP although they are ideally suited for parallel computing. Again as presented in Figure 3, SEP performs almost as well as EP, which further justifies SEP even with sampling methods. Also AEP is indistinguishable from DEP, but it reduces memory by a factor of $K$.

## C.3 Further Comparisons for SEP, DSEP and full EP

The assumption we made in the main text to achieve SEP $\approx$ full EP is that the contributions of each likelihood term to the posterior are similar. We show further results here on the approximation produced by different EP methods when there is significant heterogeneity in the data. We generated synthetic XOR classification data by sampling from 4 unit Gaussians with means $(3, 3)$, $(-3, -3)$, $(3, -3)$ and $(-3, 3)$, and labelling the clusters centered at the former two as negative examples (and positive for the others). The model $p(y_n|\boldsymbol{x}_n, \boldsymbol{\theta})$ is kernel probit regression using RBF kernel with width $l = 1.0$, which is the same as the model presented in Section 5.1 in the main text except that the features are changed to kernel representations. This makes the feature vectors high dimensional, and the local nature of kernels also makes the kernel-expanded inputs very different if the datapoints belong to different clusters. We generated $50 * 4$ test data and $\{10 * 4, 20 * 4, 50 * 4\}$ training data and ran SEP/DSEP/full EP to approximate the posterior distribution of $\boldsymbol{\theta}$. For DSEP we partitioned the dataset into 4 subsets according to the associated centroid. Each experiment was repeated 10 times to collect average test data log-likilihood and classification errors.

Table 2 shows the quatitative numbers of performances and Figure 4 visualises the contuors of probability $p(y = 1|\boldsymbol{x}, \mathcal{D})$ with true posterior approximated by $q(\boldsymbol{\theta})$. Interestingly SEP is slightly better then the others on the classification error metric. But importantly EP achieves the best test log-likelihood numbers and in general DSEP produces very similar results (shown by both the table and the figure), meaning that even for small datasets running full EP might be unnecessary. Also the three methods become indistinguishable when the size of the dataset $N$ increases. We argue the main reason is that the posterior contributions are getting similar since more datapoints are observed in the circle of kernel width.

We further tested the robustness of all three methods to outliers. We reused the settings above and randomly flipped $10\%$ labels of training data. Qualitative results in Figure 5 show that SEP is almost as robust as DSEP/EP in this example. We had tried different types of outliers and failed to find the cases where EP/DSEP significantly outperforms SEP. Future work should further characterises that when SEP gives bad approximations and separately whether it fails in the same way as EP fails, e.g. EP can fail to converge.

Table 2: Average test results of all methods on kernel Probit regression.

| $N$ | mean error | | | test log-likelihood | | |
|---|---|---|---|---|---|---|
| | **SEP** | **DSEP** | **EP** | **SEP** | **DSEP** | **EP** |
| $10*4$ | **0.032**±**0.0058** | 0.055±0.0127 | 0.032±0.0097 | -0.405±0.011 | -0.380±0.010 | **-0.378**±**0.009** |
| $20*4$ | **0.007**±**0.0014** | 0.008±0.0024 | 0.012±0.0031 | -0.326±0.007 | -0.320±0.006 | **-0.317**±**0.003** |
| $50*4$ | **0.003**±**0.0010** | 0.003±0.0014 | 0.006±0.0009 | -0.243±0.004 | **-0.233**±**0.007** | -0.238±0.003 |

(a) SEP, $N = 40$     (b) DSEP, $N = 40$     (c) EP, $N = 40$

(d) SEP, $N = 80$     (e) DSEP, $N = 80$     (f) EP, $N = 80$

(g) SEP, $N = 200$     (h) DSEP, $N = 200$     (i) EP, $N = 200$

Figure 4: Comparing predictions of kernel probit regression trained by SEP/DSEP/EP, with increasing training data size $N$.

Figure 5: Comparing predictions of kernel probit regression trained by SEP/DSEP/EP, with 10% labels flipped.

## Footnotes

[1]A little math can show the updates of alpha-EP using different existing alpha-divergence definitions are equivalent, although the corresponding alpha will change.

[2]This notation implicitly assumes that the prior and the approximate posterior belong to the approximate distribution family. In general we can propose another factor to approximate $p_0(\boldsymbol{\theta})$, and our result still applies.

[3] code adjusted from `ep-stan`: https://github.com/gelman/ep-stan