[Reviews · NeurIPS 2015]

Submitted by Assigned_Reviewer_1

QUALITY This is an interesting idea. However, I think the paper requires two major changes to be interesting to a NIPS reader.

First, the paper should be up front about its experimental nature and follow with an in-depth study. Specifically, the experiments should clearly highlight the strengths (big datasets) and weaknesses (clear, thorough comparisons and failure cases).

Second, the paper requires some editorial work. The important sections (see below) could use more explanation. The experimental study should clearly motivate the setup and carefully discuss the results with technical detail. This is what a NIPS reader would want to get from such a paper.

Some details:

Bayesian Probit Regression: I do not understand why this approximate KL is a good metric to report. Why not just compare means and variances of parameter estimates as compared to Stan/NUTS? Isn't that much more straightforward and natural?

Figure 3(c): I am confused as to why SEP with K=1 is compared to DSEP with K=10. Am I missing something here?

Mixture of Gaussians: why did you now switch to this F-norm metric? Also wouldn't it be better to simulate some data where EP fails to recover the truth? I would rather want to see whether SEP "fails in the same way" as EP, or whether the single f(\theta) factor provides a different behavior.

Having reached Section 5.3, I am a bit dissatisfied by what these first two experimental sections present. I have not learned much about the nature of SEP: specifically, I don't understand how it fails. The investigation of minibatch sizes is limited, as the experiments do not paint a clear picture. Also, given that the main motivation of this work is to scale EP, I don't understand why the majority of the experiments focus on tiny datasets.

Section 5.3 requires significant overhaul. The experimental setup is brushed aside with a citation: this is unacceptable. The results are discussed in a completely conversational manner, with little technical detail or specificity, using words like "often", "interestingly", "possibly", and "likely".

CLARITY Figure 3(b): DAEP is never discussed, either in the caption or in the text.

I am not sure what Figure 2 adds to the paper. I would rather the authors expand on Sections 3 and 4.1: both of these sections are important for the paper, and yet they would benefit from a clearer exposition. Removing Figure 2 would afford this space to clarify these important ideas of the paper.

ORIGINALITY The idea is nice. It is also original: i have not seen any work that addresses EP's memory constraints. The proposal is straightforward; thus a NIPS reader will immediately ask: "that's a good idea, I wonder why it works."

SPECIFICS line 48: ... having A myriad OF local ...

line 69: truely

line 149: summaried

line 199: arguable

line 214: this paragraph is vague. I cannot tell what you are referring to without looking at the supplementary material; thus I have little motivation to look it up in the supplementary material.

line 257: why is "mixture" capitalized?

figure 2 should appear on page 5, not page 6.

line 290: i am thoroughly confused by this comparison metric.

line 292: there is no citation numbered [25]. should it be [24]? This makes me worried about all of the citations in the rest of the paper...
Summary: This paper proposes a solution to EP's memory constraint. It does to by considering a single approximating factor instead of one per likelihood term. The paper contains little theory (and the authors admit that) but presents a slightly dissatisfactory empirical story. As such I am on the fence of accepting this paper to NIPS in its current form. Depending on the other reviews, I would encourage the authors to refresh the empirical section, clarify the narrative, and resubmit. I have no doubt that it will eventually get accepted.

Submitted by Assigned_Reviewer_2

This paper presents a new version of the EP algorithm where one iteratively refines a global approximation of the target instead of refining individual approximating factors. This reduces memory requirements and performs experimentally well compared to standard EP.

This is a useful and neat idea.

Quality: good, nice idea.

Clarity: good, however a few typos to correct.

Originality: original to the best of my knowledge.

Significance: a useful algorithm which could find quite a few applications.

Minor comments:

- The convergence of EP is not guaranteed, have the authors observed similar convergence problems for SEP?

- In the mixture of Gaussian clustering example, the posterior distribution has 4! modes. Is your SEP approximation enjoying this property? (I am not sure I fully understand Figure 4)
Summary: The paper presents a new variation of the EP algorithm where iterative refinement of a global approximation of the target is performed. Simple and interesting.

Submitted by Assigned_Reviewer_3

The paper proposes a variant of EP (SEP) which assumes the local factor (site) approximations to be identical with the result that the memory requirements are reduced by a factor of N. This is helpful in large data applications since only the global posterior approximation needs to be held in memory during the EP iterations. By experiments with probit regression, Gaussian mixture models and neural networks, the paper demonstrates that the SEP approximation can be quite close to EP in terms of accuracy.

Quality:

The proposed methods are reasonably well justified with references to existing literature and the experiments are well made. My biggest concern about the SEP approximation is the regime where the SEP/AEP assumption fails? Based on 5.1 DSEP with a manageable number of different local approximations seems to be the method of choice? Also it would be nice to illustrate possible problematic cases in terms of approximation quality with very simple examples.

Clarity:

The paper is well written and clearly structured. The analogy with SVI/VI could be explained more clearly in the main text. The remarks on the VI limit of PEP on lines 138-140 could be explained more clearly. How is this related to the discussion in [23]? Also in A.2 the approximate family and the definition of the natural parameters could be written down to clarify the text.

Lines 152-154: In many usual settings with Gaussian approximations the complexity of the site approximations is much less than D^2, and doesn't this also often depend on the number of model factors that are being approximated with each site approximation?

Originality:

The average local factor assumption is similar to AEP, which has already been published. For this reason it would be nice to summarize e.g. the DSEP/DAEP comparison from figure 2 of the supplements more clearly in the main text. Also distributed EP settings have been proposed earlier but overall I like the paper because it puts together many different variants of EP.

Significance:

Overall the paper makes incremental contributions to various directions that extend the EP framework but I like the paper because it puts together many different variants of EP.
Summary: A well written paper that makes incremental extensions to EP but puts together many different variants of EP. The models and other settings where the SEP/AEP assumption possibly fails could be illustrated more thoroughly.

Submitted by Assigned_Reviewer_4

This paper presents an elegant modification of EP that reduces memory requirements at cost of reduced accuracy (and sometimes more computation).

The paper explores connections with related algorithms.

The extensive set of experiments give a convincing argument for the practicality of the algorithm.

The only weakness of the paper is some sloppiness in the presentation.

The abstract and introduction use the word "computation" when they actually mean "memory".

Memory does not equal computation.

SEP is actually adding computation in some cases, in exchange for less memory.

The description of SEP is unclear about the number of iterations.

In Table 1, was SEP run to convergence (if such a concept exists), or was it manually stopped?

In section 4.1, it is not clear what is meant by "converges in expectation".

Exepctation of what?

With what being random?

Theorem 1 has a similar issue.

Furthermore, this claim is not proven.

The "proof" of theorem 1 in appendix A is not a proof---it is simply re-asserting the theorem.

If the authors cannot make a precise statement with a real proof then this claim should not appear in the paper.

The claim in section 4.4, that messages to latent variables associated with a single data point need not be stored, requires a special assumption.

The assumption is that the term p(x_n | h_n, theta) can be updated in one step.

This is true for the model in section 5.2, but it is not true in general.

For example, LDA has a latent variable for each document (the topic proportions) but it cannot be updated in one step.

So these messages will need to be stored, unless you want to significantly increase the amount of computation.

Appendix B.2 should say "This procedure still reduces memory by a factor of N/K".

In section 5.1, ADF should be collapsing to a delta function at the true posterior mode, not mean.
Summary: An elegant modification of EP, a nice discussion of connections to other algorithms, and extensive experimental results.

Submitted by Assigned_Reviewer_5

Summary: The paper provides a stochastic generalization of expectation propagation. In EP local contributions are removed and updated for each data point - the down side being that sufficient statistics must be saved for each data point. For models with many parameters this becomes expensive. Instead of removing the sufficient statistics for each data point, instead the authors remove the global sufficient statistic, weighted by the sample size being updated. The method is generalized to other EP variants such as parallel EP and distributed EP for when there is heterogeneity in the data set, and to models with latent variables. For the experiments the method is applied to probit regression, gaussian mixture models, and Bayesian inference in neural networks. Empirically they show that their method provides computational savings without sacrificing performance.

Quality: The method is sound, well motivated and experimentally validated in three different, but canonical, applied domains.

Clarity: Easily readable and clear. Excellent writing style.

Originality: The core idea behind the paper is simple but clever, elegantly solving the memory overhead in EP. Novel extensions of the stochastic EP method to parallel, mini-batches for better approximations, and latent variable models are also developed and explored. To my knowledge this approach and its extensions are all completely original.

Significance: The contributions are significant as they provide scalable alternatives for EP. I expect the paper to be as important to EP as SVI was to VB.
Summary: The paper is well motivated, extremely clear, and the research contributions are important for scaling EP inference for complex models with many parameters. Extensions and connections with existing EP and VB techniques are explored for an overall very thorough and mature paper.

Author Feedback
Author rebuttal: We thank the reviewers for their useful comments. A few points were unclear: these will be improved in the revision.

Exploration of failure cases:

We plan to investigate toy examples to attempt to break SEP. In our experience, across a wide number of models and datasets, SEP works almost as well as EP. When marked heterogeneity exists in data (see Fig 3b and 3c) it can be a little worse than EP and switching to DSEP with a small number of factors can help.

SEP is generally better than ADF. PBP for Bayesian NNs is the only exception where ADF can sometimes outperform SEP. The reason is that PBP uses an approximate moment matching operation which becomes more accurate as the variance of q(\theta) is shrunk to zero. This can sometimes favor ADF when a large enough number of passes over the data are used as to permit accurate moment-updates, but not so many as to overfit.

Little theory for SEP:

We are trying to address this. It is true, however, that EP also has little theory, especially when compared to SVI/VB. EP has no convergence guarantee and the "EP energy function" is not an optimization objective, but rather has the same fixed points as EP. Nevertheless, a lot of existing work has shown that EP provides in many cases accurate approximations and we expect SEP to behave also well in those cases.

Insufficient large-scale experiments:

We do report large-scale results. In the Bayesian NN example the dataset "Year" has half a million instances (see appendix, Fig 3a). Here SEP worked almost as well as EP, but EP required 65 GB of storage memory. We also believe that the experiments in 5.1 and 5.2 are also strongly indicative of the performance at scale. Arguably SEP's global approximation becomes more accurate as the number of data-points increases as it becomes less likely that a single likelihood contribution dominates the posterior. In general, it is simpler to consider smaller datasets since there it is possible to compare to EP.

Many reviewers are not worried by this aspect: R7 "...experimentally validated in three different, but canonical, applied domains"; R4 "The extensive set of experiments give a convincing argument for the practicality of the algorithm".

Since the submission we have performed further experiments on Gaussian Process classification following http://arxiv.org/pdf/1507.04513.pdf, where the largest test case has millions of data points. In all these new tests SEP works almost as well as EP.

R1:

Different metrics in Probit regression and MoG clustering:

We follow [6] and report the KL-divergence in Probit regression to summarize performance in a single number. It is hard to analytically compute the KL-divergence in the MoG example, thus we revert to the F-norm metric.

Fig 3 and mini-batches:

The reviewer probably misunderstood: K=number of approximate factors, M=number of points in parallel update (mini-batch size). Figs 3a and 3b show that the minibatch size has only a weak effect on accuracy. Fig 3c shows that having K=10 (DSEP) approximate factors rather than K=1 (SEP) leads to improved accuracy.

What Fig 2 adds to the paper:

Fig 2 plays a key role in 1) relating SEP to existing approaches and 2) visualizing the algorithmic design. Reviewers 3, 4, 5 and 7 see it as an important contribution.

R2:

In the MoGs example the posterior has 4! modes:

EP and SEP model one of the modes.

R3:

Incremental contribution:

We believe our contribution is not incremental. The paper introduces a family of approximations, DSEP, which generalizes EP and provides flexible configurations. We became aware of AEP [22], a special case of our approximation family, during preparation. Importantly, AEP has not been tested (or even considered) as a practical algorithm before, let alone the SEP and DSEP variants.

The VI limit of PEP:

Yes, we use the same techniques as in [23] to compare SVI and SEP.

Space complexity of Gaussian site approximations:

In line 152-154 we assumed Gaussian site factors with full covariance matrices, thus O(D^2) site storage. Some existing tricks (see Sec. 1, paragraph 3) still maintain N local factors and thus still scale with N.

R4:

"Convergence in expectation" and fixed point conditions:

We take the expectation over data points. We will clarify this and theorem 1 in the revision.

Retaining latent variable factors:

Thank you for pointing this out. A combination of SEP with the "Learning to pass EP messages framework" would be sensible here.

R5:

How is SEP run?:

SEP is trained using multiple passes and damping. SEP is robust to data ordering, another advantage over ADF.

Compare to SVI:

Comparisons between DSEP and SVI are important. Generally, there is little literature comparing EP to VI, but the folk wisdom is that EP is better for non-smooth potentials. In the Bayesian NN example EP-based approaches work better than SVI [4].